# Development and preliminary testing of Health Equity Across the AI Lifecycle (HEAAL): A framework for healthcare delivery organizations to mitigate the risk of AI solutions worsening health inequities

Jee Young Kim[1]*, Alifia Hasan[1], Katherine C. Kellogg[2], William Ratliff[1], Sara G. Murray[3], Harini Suresh[4], Alexandra Valladares[5], Keo Shaw[6], Danny Tobey[7], David E. Vidal[8], Mark A. Lifson[8], Manesh Patel[9], Inioluwa Deborah Raji[10], Michael Gao[1], William Knechtle[1], Linda Tang[11], Suresh Balu[1], Mark P. Sendak[1]

1 Duke Institute for Health Innovation, Duke Health, Durham, North Carolina, United States of America, 2 Sloan School of Management, Massachusetts Institute of Technology, Cambridge, Massachusetts, United States of America, 3 Division of Hospital Medicine, University of California San Francisco, San Francisco, California, United States of America, 4 Cornell University, New York, New York, United States of America, 5 Community representative, Durham, North Carolina, United States of America, 6 FDA Regulatory Group, DLA Piper, San Francisco, California, United States of America, 7 AI and Data Analytics, DLA Piper, Dallas, Texas, United States of America, 8 Center for Digital Health, Mayo Clinic, Rochester, Minnesota, United States of America, 9 Division of Cardiology, Duke Health, Durham, North Carolina, United States of America, 10 Department of Electrical Engineering and Computer Science, University of California Berkeley, Berkeley, California, United States of America, 11 School of Medicine, Johns Hopkins University, Baltimore, Maryland, United States of America

* jee.young.kim@duke.edu

## Abstract

The use of data-driven technologies such as Artificial Intelligence (AI) and Machine Learning (ML) is growing in healthcare. However, the proliferation of healthcare AI tools has outpaced regulatory frameworks, accountability measures, and governance standards to ensure safe, effective, and equitable use. To address these gaps and tackle a common challenge faced by healthcare delivery organizations, a case-based workshop was organized, and a framework was developed to evaluate the potential impact of implementing an AI solution on health equity. The Health Equity Across the AI Lifecycle (HEAAL) is co-designed with extensive engagement of clinical, operational, technical, and regulatory leaders across healthcare delivery organizations and ecosystem partners in the US. It assesses 5 equity assessment domains–accountability, fairness, fitness for purpose, reliability and validity, and transparency–across the span of eight key decision points in the AI adoption lifecycle. It is a process-oriented framework containing 37 step-by-step procedures for evaluating an existing AI solution and 34 procedures for evaluating a new AI solution in total. Within each procedure, it identifies relevant key stakeholders and data sources used to conduct the procedure. HEAAL guides how healthcare delivery organizations may mitigate the potential risk of AI solutions worsening health

**Data Availability Statement:** No quantitative data were used or generated for this study.

**Funding:** This work was supported by the Gordon and Betty Moore Foundation (Grant # 10849 to JYK, AH, SM, HS, AV, DEV, MAL, MP, IDR, and MPS). The funders had no role in study design, data collection and analysis, decision to publish, or preparation of the manuscript.

**Competing interests:** WR, MG, SB, and MPS have declared co-inventing software at Duke University licensed by Duke University to external commercial entities Clinetic, Cohere Med, Kela Health, and Fullsteam Health. MG, SB, and MPS also own equity in Clinetic. No other competing interests were declared.

inequities. It also informs how much resources and support are required to assess the potential impact of AI solutions on health inequities.

## Author summary

In healthcare, the use of data-driven technologies such as Artificial Intelligence (AI) and Machine Learning (ML) is increasing. However, the lack of robust regulations and standards poses a challenge to their safe and equitable use. To bridge this gap, we brought together healthcare leaders from various backgrounds in a workshop and developed the Health Equity Across the AI Lifecycle (HEAAL) framework. HEAAL evaluates how the use of AI might affect health equity. It examines five crucial domains—accountability, fairness, fitness for purpose, reliability and validity, and transparency—across eight key decision points in the AI adoption process. HEAAL offers tailored procedures for assessing both existing and new AI solutions, along with relevant stakeholders and data sources. By providing step-by-step guidance, HEAAL empowers healthcare delivery organizations to comprehend and mitigate the risk of AI exacerbating health inequities.

## Introduction

The use of data-driven technologies such as Artificial Intelligence (AI) and Machine Learning (ML) is growing in healthcare. These technologies can be valuable tools for streamlining clinical workflow, aiding clinical decision-making, and improving clinical operations [1–4]. For example, the integration of AI and ML in healthcare helps in the detection and management of sepsis [5], preventing unanticipated intensive care unit transfers [6], and automated calculation of left ventricular ejection fraction [7]. AI and ML can promote earlier detection of diseases, more consistent collection and analysis of medical data, and greater access to care [8].

However, the proliferation of healthcare AI tools has outpaced regulatory frameworks, accountability measures, and governance standards to ensure safe, effective, and equitable use [3,9,10]. Past research has shown numerous incidents where healthcare AI technologies perpetuate bias and inequities [11–13]. To address this issue, in 2022 and 2023, government officials from the White House [14], HHS Office of Civil Rights [15], Office of the National Coordinator for Health Information Technology (ONC) [16], and Office of the Attorney General in California [17] took action to protect against healthcare AI worsening inequities. While these regulatory actions describe what harms to avoid, they also leave significant room for interpretation of how healthcare delivery organizations can implement these principles.

Numerous academic papers have surfaced potential causes of bias in AI products, including lack of representation and diversity in model training data [18–20], lack of sufficient historic data to build an accurate model [21], an outlier event with unprecedented data [22], bias captured in specific data measurements [23,24], bias captured in unstructured text [25,26], bias embedded within outcome labels used to train models [11,12], and models learning shortcuts unrelated to disease process to generate diagnostic predictions [27,28]. Numerous reviews and frameworks have described categories of bias in AI products and proposed steps to address them [29–34]. But to date, there has yet to be a comprehensive set of actionable procedures across the AI product lifecycle for healthcare delivery organization leaders to adopt internally to mitigate the risk of AI products worsening health inequities.

Our prior work revealed that healthcare delivery leaders find it challenging to identify and objectively measure the potential impact of an AI product on health inequities. We interviewed 89 individuals from 10 US healthcare delivery organizations and ecosystem partners [35]. Even though we interviewed 13 AI ethics and bias experts, we were not able to reach a consensus on the best approaches to assess AI products for potential impacts on health inequities.

### Present research

To address these gaps and tackle a common challenge faced by healthcare delivery organization leaders, we, the Health AI Partnership (HAIP), organized a case-based workshop [36] and developed a framework to assess how the use of AI solution might affect health equity. In the present research, we define health equity as *the attainment of the optimal health for all people regardless of race, ethnicity, disability, sexual orientation, gender identity, socioeconomic status, geography, preferred language, and other factors that may affect access to care and health outcomes* [37]. The manuscript offers a comprehensive overview of the development and testing of the framework designed specifically for leaders in healthcare delivery organizations. We named this framework Health Equity Across the AI Lifecycle (HEAAL). We aim to (1) provide a detailed overview of the procedures in the framework, along with relevant data sources and stakeholders and (2) describe in detail the participatory design research methodologies used to develop the framework to inform future stakeholder engagement efforts.

## Materials and methods

### Engage and align

HEAAL was collaboratively designed through extensive engagement with clinical, operational, technical, and regulatory leaders from healthcare delivery organizations and ecosystem partners in the US (Fig 1). Three innovation teams were recruited as case study teams. They curated case studies and presented them at the workshop. Seventy-seven representatives from ten healthcare delivery organizations and four ecosystem partners participated in the workshop and shared their experiences in adopting AI within their respective settings. Six framework developers—a clinician, a community representative, a computer scientist, a legal and regulatory expert, a project manager, and a sociotechnical scholar—were recruited to create a scaffolding of the framework and develop its procedures. Eight HAIP leaders who have clinical, community engagement, computer science, operational, and regulatory expertise evaluated the framework and provided feedback. Three design researchers facilitated the framework design process by collecting and synthesizing data from all other participants, refraining from generating data themselves. Design involved two rounds of divergent and convergent processes with four phases: discover, define, develop, and deliver (Fig 2).

### Ethics statement

The present research was considered a quality improvement (QI) project that did not involve human subjects research. Thus, it was exempted from IRB review and approval at Duke University Health System. All participants provided verbal consent to participate in the co-design processes and to have anonymized data used in analyses.

### Discover

During the Discover phase, the problem was widely explored by speaking to all participants and documenting their responses.

| Participant | | Role | Responsibilities |
|---|---|---|---|
| (C) | Case study presenters | 3 innovation teams that develop and implement AI solutions in healthcare delivery organizations | Curated a case study, presented it at the workshop, and tested out the framework |
| (W) | Workshop participants | 77 stakeholders from 10 healthcare delivery organizations and 4 ecosystem partners with clinical, technical, operational, regulatory, and AI ethics expertise | Contributed to developing the procedures of the framework |
| (F) | Framework developers | A clinician, a community representative, a computer scientist, a legal and regulatory expert, a project manager, and a sociotechnical scholar | Created a scaffolding of the framework and contributed to developing its procedures |
| (H) | HAIP leaders | A clinical data scientist, a community organizer, 2 computer scientists, 3 lawyers, and a program director | Evaluated the framework and provided feedback |
| (D) | Design researchers | A clinical data scientist, a project manager, and a qualitative research scientist | Facilitated the co-design process by collecting, iterating, and synthesizing data from all other participants |

**Fig 1. Participants and their roles and responsibilities in co-designing HEAAL.**

## Curate case studies

A total of three case studies were curated. A Duke Institute for Health Innovation (DIHI) team developed an initial example case study for a pediatric sepsis prediction algorithm. This case study was not presented at the workshop but was used to illustrate the case study format to other teams. Teams from NewYork-Presbyterian (NYP) and Parkland Center for Clinical Innovation (PCCI) then curated case studies for postpartum depression and patient segmentation algorithms, respectively, using the structure provided by the DIHI team [38,39]. The case

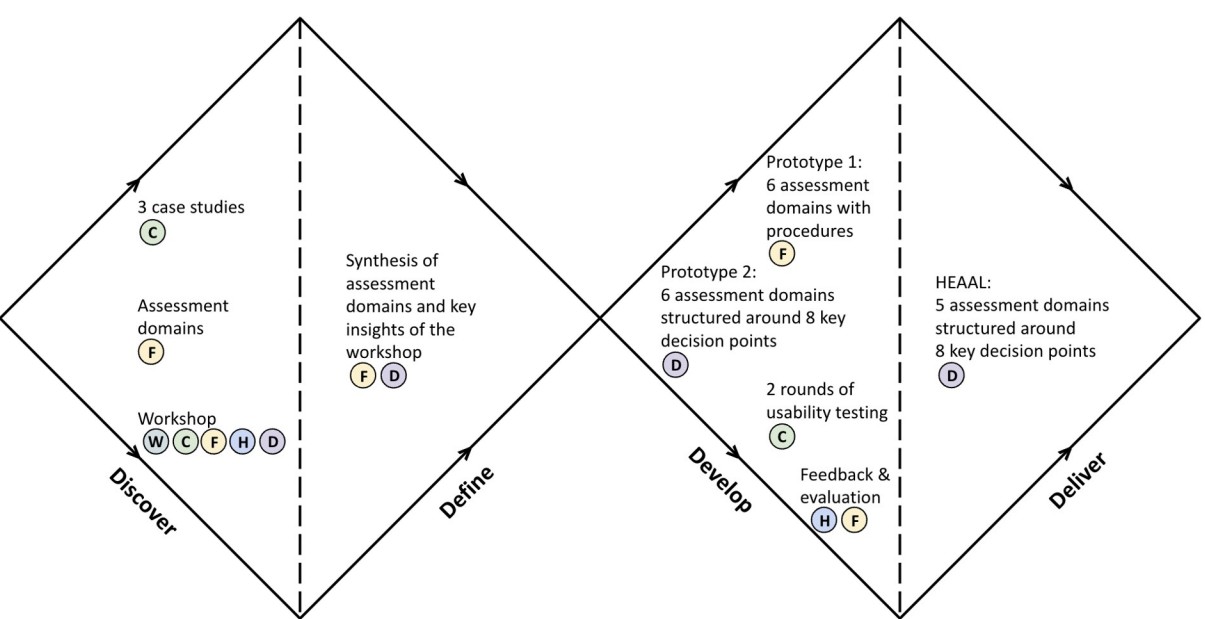

**Fig 2. Four phases of co-design processes and participants engaged in each phase.**

studies served as real-world examples to facilitate ideation and discussion during the workshop among participants. More information about the workshop is presented in the accompanying Formal Comment [36].

### Surface domains of assessment

Six framework developers individually reviewed two case studies and were asked to identify major domains of assessment or concerns that healthcare delivery organization leaders should assess when deciding to implement an AI solution into clinical practice safely, effectively, and equitably. For each domain of assessment, they were asked to provide its descriptions and propose how each domain may be assessed and what data may be required. Design researchers compiled responses from all framework developers in a single place and made the framework developers cluster similar ideas together. Ultimately, this activity resulted in the creation of eight unique clusters.

### Surface novel insights from the workshop

Seventy-seven people with various domains of expertise from 10 healthcare delivery organizations and 4 ecosystem partners attended the workshop. Clinical, technical, operational, and regulatory stakeholders as well as AI ethics experts shared their perspectives on the workshop topic through different activities as described in the accompanying Formal Comment [36]. Design researchers took notes of the discussions that took place during the workshop.

### Define

During the Define phase, responses collected from all participants were synthesized.

### Synthesize key insights of the workshop

After the workshop, the design researchers reviewed and extracted key insights from their notes that they took during the workshop. Then, the design researchers mapped them onto the eight clusters of assessment domains that the framework developers had previously created. This activity ensured that novel ideas shared by the workshop participants were incorporated into the framework's content.

### Synthesize domains of assessment

Framework developers and design researchers thoroughly reviewed the updated eight clusters of assessment domains and merged clusters with similar ideas. This process resulted in six domains of assessment with relevant guiding questions.

### Develop

During the Develop phase, prototypes were developed and tested.

### Generate the first prototype

Framework developers individually provided answers to each guiding question listed under the six domains of assessment. Design researchers consolidated responses from all framework developers into a single document, organizing them sequentially to serve as procedures for assessing concerns described in each guiding question. Framework developers and design researchers then iterated on the document together. The revised document became the first prototype for the framework. It contained six assessment domains and relevant sets of actionable procedures under each of the assessment domains.

## Conduct initial usability testing

Data scientists from the DIHI case study team tested the first prototype of the framework by applying its procedures to analyze a pediatric sepsis prediction algorithm. This process was essential to ensure that the framework was pragmatic and usable in practice. After the analysis, they reported the results of the analysis, shared their experiences using the framework, and suggested areas of improvement.

A major suggestion that the data scientists proposed was to consider restructuring the framework. They found that some of the procedures were redundant across different assessment domains. Such redundant procedures created inefficiencies, making data scientists go back and forth between different assessment domains to repeat similar analyses. To address this issue, they recommended listing the procedures of all assessment domains sequentially using the previously developed HAIP eight key decision points of the AI product life cycle [35].

Another suggestion that the data scientists provided was to describe some of the procedures more concretely with actionable guidance. For example, the data scientists requested the framework to explicitly state the required personnel or resources for each procedure. Similarly, they requested more detailed descriptions of the roles and responsibilities of individual decision-makers, advocating for statements like "seek approval from ______ stakeholder" instead of "engage ______ stakeholder."

## Generate the second prototype

By incorporating feedback from the data scientists, design researchers generated the second prototype of the framework. The second prototype mapped procedures from the 6 domains of assessment to the HAIP eight key decision points of the AI product life cycle [35]. At this stage, tags were added to each procedure for relevant stakeholders to be involved, relevant datasets required for analyses, and health equity assessment domains.

## Conduct advanced usability testing

A project manager and two data scientists from the DIHI case study team were recruited to test the usability of the second prototype. The team followed the procedures described in the framework to analyze the same pediatric sepsis prediction algorithm. With the updated content and structure of the framework, it was important to examine whether the framework addressed the initial pain points raised from the initial usability testing.

The case study team was atisfyied with the updated structure of the framework. They liked how the procedures flowed sequentially from the beginning to the end of the AI lifecycle. The project manager reported that while the framework demanded substantial effort, it remained manageable to navigate. The project manager found the framework to be particularly helpful in understanding potential gaps in algorithms. The data scientists provided additional feedback on how the assessment could be conducted more efficiently. They suggested rearranging some of the procedures in a different sequential order and modified descriptions of some procedures. They also suggested that once each procedure is completed, users should understand how to interpret the outputs of the procedure and what to do next.

## Seek general feedback and evaluation

The second prototype was also shared with the framework developers and the HAIP leadership team for review. One major concern was that the framework does not sufficiently describe procedures related to one of the assessment domains, "policy and regulation." HAIP leaders with

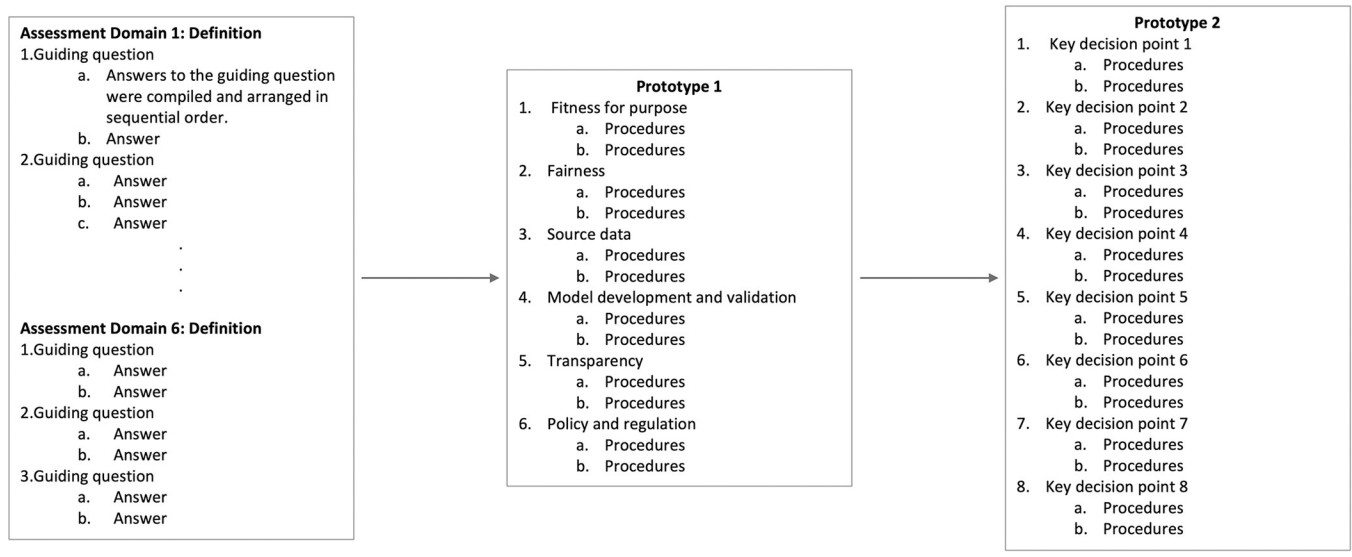

**Fig 3. Prototype development during the Develop phase.** Responses to guiding questions were gathered and synthesized to create the initial prototype. It contained procedures for evaluating six health equity assessment domains. After the initial testing by a case study team, this prototype evolved into the second prototype. The second prototype was structured around eight key decision points of AI adoption and tested by the case study team. It was then shared with the framework developers and the HAIP leadership team for feedback and evaluation.

regulatory expertise cautioned that engaging regulatory stakeholders in some procedures was not sufficient to assess the policy and regulation domain. Fig 3 shows how prototypes were developed during the Develop phase.

## Deliver

During the Deliver phase, the final prototype was refined and prepared for dissemination. Design researchers incorporated all feedback into revising the prototype and generated the first version of the framework. The framework was named Health Equity Across the AI Lifecycle (HEAAL). HEAAL was then shared with two other case study teams from NYP and PCCI. They plan to apply HEAAL in evaluating their postpartum depression and patient segmentation algorithms and publish their findings.

## Results

HEAAL, presented in the supporting information (S1 Appendix and S2 Appendix), was established after conducting a series of activities, including curating case studies, surfacing domains of assessment, hosting a workshop, synthesizing insights, developing two prototypes, conducting two rounds of usability testing, and gathering feedback. Over the course of seven months, clinical, technical, operational, and regulatory stakeholders and AI ethics experts from healthcare delivery organizations and ecosystem partners contributed a great amount of their time and effort to these framework development activities.

### Five domains of assessment

HEAAL addresses five health equity assessment domains. The five equity assessment domains are (1) accountability, (2) fairness, (3) fitness for purpose, (4) reliability and validity, and (5) transparency.

Accountability refers to the principle of holding individuals, organizations, or systems responsible for their actions, decisions, and outcomes of the proposed AI solution. This assessment domain entails overseeing potential substantial adverse impacts that may arise after the solution is integrated, identifying stakeholders responsible for managing and controlling the solution throughout its lifecycle, and developing plans for continuous monitoring. It highlights the role of a governance committee or designated stakeholders within a healthcare delivery organization who may oversee the risk of potential negative consequences arising from solution use. It suggests that the governance committee or designated stakeholders should have a clear understanding of the legal and internal policy constraints that the solution is subject to comply and proactively develop intervention plans. Additionally, they should devise strategies for ongoing monitoring, feedback, and evaluation. The assessment of accountability ensures that the solution remains adaptable to evolving circumstances and emerging health equity concerns, sustains safe performance and continues to improve over time.

Fairness is defined as the ethical principle of treating individuals or groups impartially and without bias in the procurement, development, integration, and maintenance of the proposed AI solution. This assessment domain focuses on equal allocation of resources and opportunities across different individuals or groups to prevent any unjust or discriminatory outcomes that may arise from the use of the solution. It involves establishing and evaluating fairness criteria for the model performance and its work environment. The assessment of fairness ensures that the solution performs equitably across disadvantaged and advantaged patient subgroups and helps healthcare delivery organizations track progress towards achieving equity objectives. By understanding factors that contribute to potential inequitable technical, clinical, and operational outcomes, fairness assessment strives to mitigate existing disparities and prevent new ones that may arise from the adoption of the solution.

Fitness for purpose is defined as the extent to which the proposed AI solution is appropriate for solving the identified problem posed by the intended use. This assessment domain evaluates whether the solution aligns with the specific goals, requirements, and contexts for which it was designed and implemented. It involves defining the intended and unintended use, constraints, and the target population for the solution. It also encompasses evaluating the suitability of a ML model compared to a simpler heuristic model for addressing the problem at hand. The fitness for purpose assessment emphasizes the engagement of its intended users and patient community members from the target population in the evaluation process. The active involvement ensures that the solution aligns not only with technical specifications but also with the broader goals and needs of its intended users, patient community members, and other relevant stakeholders within a specific context. Ultimately, the fitness for purpose assessment ensures that the solution is designed to address the identified problem comprehensively across disadvantaged and advantaged patient subgroups.

Reliability and validity refer to the performance of the proposed AI solution regarding its consistency and accuracy. A reliable model produces consistent and reproducible output with the same input or similar data over multiple instances. Reliability promotes confidence in the solution's performance. A valid model presents output that accurately measures or predicts the intended outcome of interest. Validity ensures that the model is measuring what is intended to measure, reflecting the real-world phenomenon it is meant to represent, and addressing the specific problem it was designed for. The assessment of reliability and validity ensures that the solution consistently achieves pre-specified performance targets across technical and clinical measures.

Transparency is defined as the clarity and openness to explain how the proposed AI solution is developed, integrated, and maintained. This assessment domain highlights the importance of comprehensive communication with users and other affected stakeholders, including

members of disadvantaged and advantaged patient subgroups. Effective communication should go beyond providing details about the technical specifications of the model and its intended use. It should entail the disclosure of information related to potential harms, risks, limitations, and impacts associated with the solution. The assessment of transparency empowers users and other affected stakeholders to make informed decisions in using the solution and helps them make progress towards equity objectives.

Initially, "policy and regulation" emerged as one of the health equity assessment domains. Throughout the entire co-design process, participants expressed the importance of healthcare delivery organizations adapting to the changing regulatory landscape. However, ultimately it was not included in HEAAL, as there was no universal set of procedures that applied to diverse AI use cases across the US. Given the dynamic nature of regulations, the broad coverage of health equity assessment concerns within the framework, and the large number of jurisdiction-specific actions, HAIP leaders confirmed that no single set of procedures could adequately address policy and regulation across diverse AI use cases. For the time being, healthcare delivery organizations need to monitor federal and local regulators, including offices of state Attorney Generals and departments of health. A forum for streamlining and summarizing the evolving landscape may be needed so that healthcare delivery organizations have a go-to place to ensure that they comply with federal and local policy and regulation. New procedures may need to be added to HEAAL to support healthcare delivery organizations seeking to comply with emerging regulations and policies.

## Structure and procedures

HEAAL is a process-oriented framework that spans across eight key decision points of the AI lifecycle. The key decision points framework was adopted because of its practicality for healthcare delivery organization leaders. Our previous work found that health system leaders find the key decision points framework useful in practice because it aligns with their approach to technology adoption [35]. The eight key decision points encompass decisions made not only within the technical domain but also across strategic, operational, and clinical domains during the adoption of AI in healthcare. Thus, the adoption of the key decision points framework facilitated the evaluation of not just the technical aspects of the AI product but also its work environment, encompassing end users, clinical workflow, and business strategies in relation to the five health equity assessment domains.

HEAAL contains 37 procedures for evaluating an existing AI solution and 34 procedures for evaluating a new AI solution. When evaluating an existing solution, additional procedures are required in the second and fifth decision points to make sure that the solution aligns with the implementation context. While all standard procedures should be applied to all AI solutions of interest, some procedures are tested at different decision points or follow a different sequential order based on whether the solutions already exist or not. For example, the set of procedures for testing an existing solution conducted in the second decision point is deferred until the fifth decision point when healthcare delivery organizations develop a new solution. To differentiate between two scenarios, procedures for evaluating an existing AI solution are written in red and black text, while procedures for assessing a new AI solution are presented in blue and black text within the framework (S1 Appendix and S2 Appendix).

Each procedure not only contains sub-procedures with detailed steps but also identifies relevant active stakeholders and data sources. Across HEAAL procedures, eight different types of stakeholders (Table 1) are involved, and six different types of data (Table 2) are used for assessing the impact of a new AI solution on health equity. Active stakeholders–other than the product manager and clinical champion–are listed for each procedure. Product managers and

**Table 1. Stakeholders involved in completing the HEAAL procedures.**

| Stakeholder type | Definition | Example roles |
|---|---|---|
| Strategic (S) | Stakeholders who develop strategic plans and make decisions that align with organizational interests | Senior leaders (e.g., CEO, CMO), departmental leaders (e.g., medical directors) |
| Operational (O) | Stakeholders who manage workflow and make decisions to integrate | Business unit leaders (e.g., nursing supervisors), diversity, equity, and inclusion (DEI) roles, frontline workers |
| Clinical (C) | Stakeholders who provide clinical care to patients | Frontline clinicians, end-users |
| Technical (T) | Stakeholders who develop the model and its infrastructure | Data scientists, data engineers, UI/UX designers, health IT |
| Regulatory (R) | Stakeholders who review the model from regulatory, compliance, and ethical perspectives | Legal, regulatory affairs, local governance committee, IRB |
| Patient (P) | Stakeholders who receive clinical care and provide insights on their experiences | Patients, patient community representatives |
| Clinical champion | Clinical stakeholders who lead the project and provide clinical expertise in model development | |
| Project manager | Stakeholders who manage the project and communicate with various stakeholders involved in the project | |

clinical champions are assumed to be part of the entire AI solution lifecycle and thus, are not explicitly listed in every procedure. Completing the procedures in each key decision point involves various stakeholders and data sources and ensures that a selected AI solution is evaluated against five assessment domains for health equity. Fig 4 provides the overview of HEAAL. It outlines health equity assessment domains, active stakeholders, data sources, and testing highlights at each key decision point.

**Table 2. Sources of data used to complete the HEAAL procedures.**

| Data source | Definition |
|---|---|
| Local healthcare retrospective data | Historical healthcare data that is curated within the primary healthcare delivery organization seeking to adopt an AI product. The local data can be sourced from a variety of systems, including the EHR, radiology PACS system, medical claims, audit logs, electrocardiograms, and high-frequency vital sign monitors. When a model is internally developed, the local healthcare retrospective data set is used for training the model. |
| Local healthcare prospective data | Real-time healthcare data that is curated within the primary healthcare delivery organization seeking to adopt an AI product. The local data can be sourced from a variety of systems, including the EHR, radiology PACS system, medical claims, audit logs, electrocardiograms, and high-frequency vital sign monitors. The local healthcare prospective data set is used for validating a model during a 'silent trial' and for using the model in clinical care. |
| Local non-healthcare data | Non-healthcare data that is curated within a geographic setting where a healthcare delivery organization is based. The local non-healthcare data can be derived from a variety of external sources, including US Census. |
| Training data | Data used for training a model. When the model is externally developed, the training data set contains data from an external source. When the model is internally developed, the training data set is sourced from local healthcare retrospective data. |
| Literature review | Data collected through reviewing previously published scholarly works on a specific topic. |
| Organizational data | Data that describes characteristics of organizations, their internal structures, processes, and behavior as corporate actors in different social and economic contexts. The organizational data includes Key performance Indicators (KPIs) that quantify progress toward strategic and operational goals. |
| Qualitative data | Data collected through qualitative research methods, including surveys, focus groups, and interviews. |

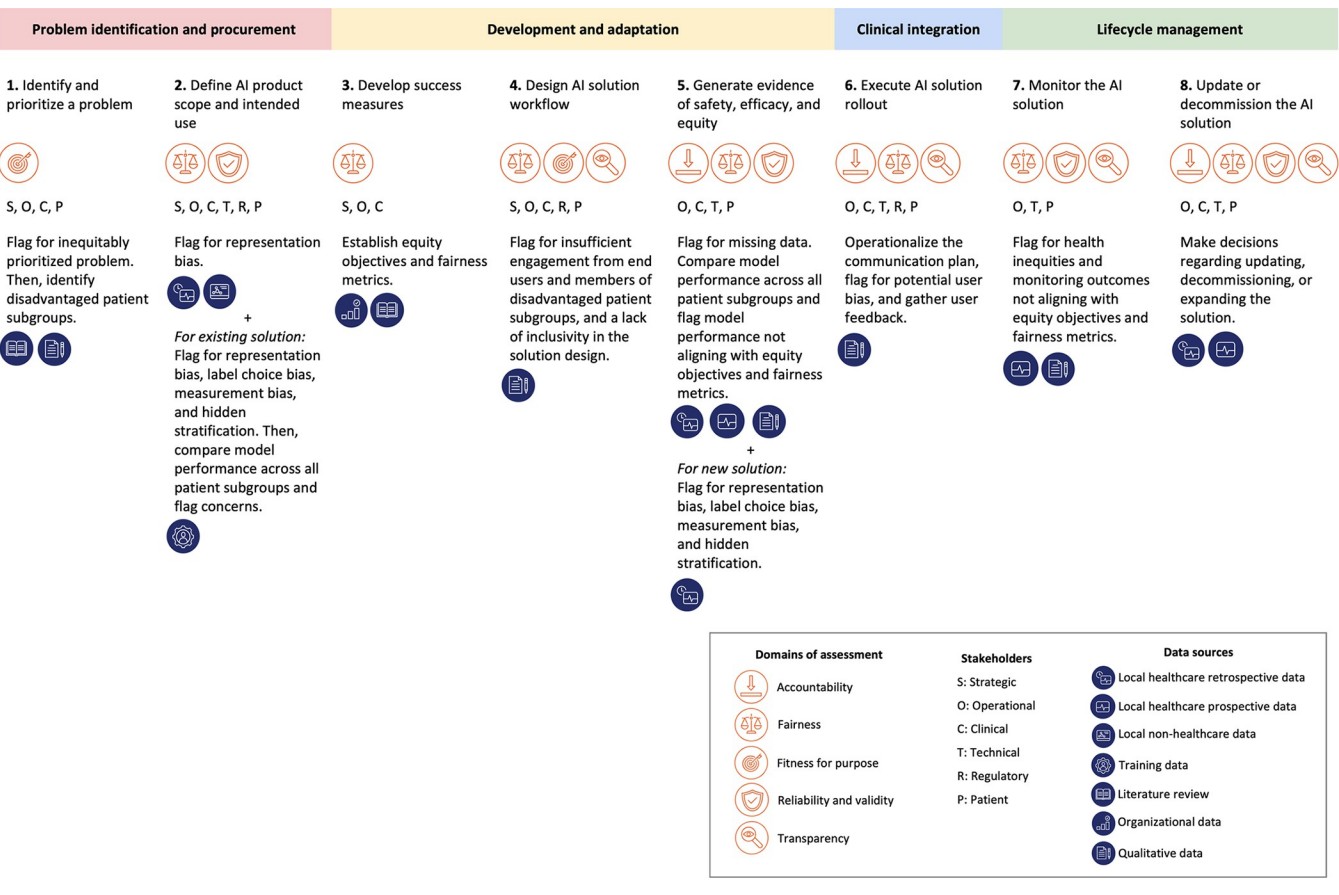

**Fig 4. Overview of HEAAL.** HEAAL delineates health equity assessment domains, active stakeholders, data sources, and testing highlights across eight key decision points.

## Achieving health equity

HEAAL provides guidance for healthcare delivery organizations to assess the baseline level of health inequity, establish equity objectives for implementing the chosen AI solution, and evaluate the progress towards these objectives across eight key decision points in the AI lifecycle. The assessment of the baseline level of health inequity involves several procedures during the initial two decision points. First, it begins with an analysis of the current state of health inequity through conducting a literature review on epidemiology and consulting with personnel who have a deep understanding of patient experiences, such as healthcare providers, patient navigators, and patient community members. Then, local healthcare retrospective data is scrutinized to determine the presence of identified health inequities within the local healthcare delivery setting. The information obtained from both procedures is synthesized to compile a comprehensive list of health inequities and to identify disadvantaged patient subgroups.

Following the measurement of the current state of health inequity, the third decision point entails establishing equity objectives for implementing the AI solution in terms of both health and economic outcomes. These objectives may span from maintaining the current level of inequity to reducing it significantly. Defining the equity objectives involves identifying the most suitable fairness metrics for the AI product to attain the established goals. Additionally, it requires documenting the rationale behind the selection of these specific fairness metrics.

The pursuit of equity objectives progresses through the subsequent three decision points. The fourth decision point centers on solution design. Solution design is informed by input from both end-users and members of disadvantaged patient subgroups. By engaging end-users, the solution becomes accessible, inclusive, and usable by all. Involving members of disadvantaged patient subgroups uncovers specific support needs to ensure they derive maximum benefit from the solution. At the fifth decision point, there is a strong emphasis on evaluating the performance of the model using both retrospective and prospective data sourced from local healthcare providers. This assessment entails conducting a thorough evaluation of the model performance against fairness metrics across disadvantaged and advantaged patient subgroups. At the sixth decision point, the focus is on communication and education provided to end-users, members of disadvantaged patient subgroups, and other stakeholders affected by the clinical integration of the AI solution. This outreach raises awareness about existing health inequities, potential biases among users, and their consequences. Moreover, it facilitates the collection of feedback, ultimately advancing progress towards equity objectives.

The final two decision points involve ongoing monitoring of shifts in health inequities among disadvantaged and advantaged patient subgroups. This continuous evaluation determines whether the implementation of the AI solution moves the organization closer to its equity objectives. If the monitoring results diverge from these objectives, the AI solution undergoes either updates or decommissioning.

## Decoupling algorithmic fairness from health equity

HEAAL includes procedures that focus on components of the AI model, including training data and outcome labels, and components of the implementation context, including personnel availability and resources for lifecycle management. Procedures that focus on algorithmic fairness are distinct from those that focus on potential impact on health equity. This allows for scenarios that may initially seem unintuitive, in which algorithmic fairness and health equity impacts do not align. For example, consider the scenarios in Table 3.

Scenarios A and D are consistent with the dominant narrative that closely couples algorithmic fairness and impacts on health equity. In scenario A, an AI solution performs well on a disadvantaged subgroup and once integrated into clinical care enables progress towards an equity objective to improve outcomes for that disadvantaged subgroup. Conversely, in scenario D, an AI solution performs poorly on a disadvantaged subgroup and once integrated into clinical care further widens a health inequity for that disadvantaged subgroup.

Awareness of scenario B is increasing. In one published case study, an AI product built to identify patients at high risk of missing appointments was assessed for use in patient scheduling. A workflow to use the algorithm to double-book patients at high risk of no-shows was determined to worsen health inequities [40]. In other scenarios, an AI product with strong performance across both disadvantaged and advantaged subgroups may be integrated into a healthcare delivery organization in which resources and personnel are unequally distributed.

Table 3. Four scenarios of alignment between algorithmic fairness and health equity.

|  | AI solution advances health equity | AI solution fails to advance health equity |
|---|---|---|
| AI solution addresses all algorithmic fairness concerns on historical data | Scenario A (Aligned: Fair algorithm promoting health equity) | Scenario B (Unaligned: Fair algorithm worsening health equity) |
| AI solution fails to address algorithmic fairness concerns on historical data | Scenario C (Unaligned: Unfair algorithm promoting health equity) | Scenario D (Aligned: Unfair algorithm worsening health equity) |

Under-resourced settings that care for disadvantaged subgroups may not be able to allocate the same level of personnel effort as higher-resourced settings to follow up on AI model outputs. Prospective implementation of the AI solution that performs well on measures of fairness could maintain or worsen health inequities.

Lastly, scenario C goes against the dominant narrative of AI. The framework development process surfaced at least two categories of use cases in scenario C. In both categories, there is an inequity in the workup or diagnosis of a medical condition targeted by the AI solution. In the first category of scenario C, which we call "inequitable underdiagnosis," the medical condition is evenly distributed across advantaged and disadvantaged subgroups. Due to inequities in workup or diagnosis, the medical condition is underdiagnosed in disadvantaged subgroups. Example use cases within this category include AI products that target peripheral artery disease (PAD), chronic kidney disease (CKD), and mental illness. An AI solution may appear to perform poorly on historical data for a disadvantaged subgroup compared to an advantaged subgroup. However, estimates of model performance on historical data are inaccurate because a substantial portion of positive cases (e.g., patients with PAD, CKD, or mental illness) in the disadvantaged subgroup are undiagnosed. Prospective implementation of the AI solution with proactive outreach to conduct appropriate workup and diagnosis for all high- risk patients will be required to assess the impact on health equity.

In the second category of scenario C, which we call "inequitable overdiagnosis," the medical condition is unevenly distributed across advantaged and disadvantaged subgroups. Due to inequities in workup or diagnosis, the medical condition is over-diagnosed in disadvantaged subgroups. Example use cases within this category include behavioral emergencies in the inpatient setting that can prompt the use of physical or chemical restraints, child abuse or neglect that can prompt family separation, and organ transplant ineligibility. An AI solution may appear to perform poorly (or better) on historical data for a disadvantaged subgroup compared to an advantaged subgroup. However, systemic racism may be entangled in the diagnosis process and the equity objective can be to reduce event rates across both disadvantaged and advantaged subgroups. Prospective implementation of the AI solution with proactive outreach to provide medical and social support for all high-risk patients can improve health equity.

## Discussion

Healthcare delivery organizations are grappling with how to ensure that AI does not worsen health inequities. To mitigate the risk of AI worsening health inequities, a community of clinical, technical, and operational leaders within healthcare delivery organizations convened to strengthen internal AI governance programs. Through developing and testing the HEAAL framework, we provide healthcare delivery organizations with actionable guidance on how to approach this challenge. Below, we describe how the HEAAL framework is differentiated from prior work and makes a unique contribution to the field.

### Community-generated framework

HEAAL is a community-generated framework. Stakeholders across healthcare delivery organizations and relevant domains of expertise, including community engagement, were actively engaged and their concerns were systematically captured through a rigorous co-design process. We used a case-based workshop method to ground the initial discovery activities. This approach helped us create a comprehensive framework for equity assessment by gaining broad input from a diverse community of practitioners. An important advantage of this method is that it can promote honest discussions of bold and diverse ideas on a sensitive subject while establishing trust and safety among those involved.

Another strength of this method is its use of real-world examples. The use of real-world examples made it easier for participants to relate to the work presented and unpack complex concepts. As a result, all discussions and recommendations for HEAAL are grounded in the experiences of practitioners who implement and evaluate similar solutions in their institutions.

## Comprehensive and usable framework

HEAAL procedures are designed to be comprehensive. It contains a comprehensive set of procedures that are tailored to new and existing AI solutions and span all stages of the AI adoption lifecycle. Comprehensive procedures mitigate ambiguity when evaluating the impact of a new AI solution on health equity across the entire lifecycle of an AI solution. Mutually exclusive procedures ensure that there is no redundancy across procedures and that no single procedure outweighs others.

By conducting multiple rounds of usability testing that applied the framework to a real use case, we ensured that the procedures were clearly written and usable in practice. Every procedure contains step-by-step guidance to support users.

## Implications for practice

The HEAAL framework highlights four complex challenges that will require significant attention and investment by diverse stakeholders. First, the framework exposes an Achilles heel of AI by emphasizing the role of context-specific factors in health equity assessments. AI solutions are portrayed as highly scalable and able to rapidly deliver value to healthcare organizations. This perception has gained significant momentum since the emergence of Large Language Models (LLMs). However, the HEAAL framework is applied in a context-specific fashion that is not easily scalable. An AI solution that is evaluated by one setting through HEAAL should be reassessed when a different setting considers implementation. Even if the same technology is being used, the assessment of the technology could reach different conclusions when the setting changes. Different contexts in the use case involve variations in patient population, stakeholders, sources of data, and clinical workflow. For example, application of the framework in one context could suggest that an AI tool will advance equity, while application of the framework in a different context could suggest that the same AI tool will worsen inequities. Thus, to ensure health equity, HEAAL should be applied every time a healthcare organization considers using an AI solution.

Second, successful implementation of HEAAL requires significant expertise, technology infrastructure to gather diverse robust datasets, and personnel effort. Despite the framework being publicly accessible and consensus among healthcare leaders to eliminate bias in AI, healthcare delivery organizations will not be able to apply the entire framework to every AI solution without significant support. HEAAL emphasizes the importance of collaborative governance models for medical AI, in which centralized authorities (e.g., FDA, CMS) coordinate and support local governance activities [41]. Significant infrastructure and technical assistance investments must be made to support low-resource settings to adopt HEAAL.

Third, applying a tool like HEAAL must be accounted for in reimbursement for medical AI. An AI procurement and implementation process that uses HEAAL will necessitate higher investment than a process that skips the assessment of health equity impacts. Without financial incentives to support the adoption of HEAAL, healthcare delivery organizations seeking to minimize discrimination due to AI will avoid AI products altogether, even if an AI solution could improve quality, safety, and equity. One potential financial incentive is to reimburse products that advance equity objectives through a rigorous HEAAL assessment at a higher rate than products lacking such evidence.

Fourth, there is concern that HEAAL can serve as a 'rubber stamp' for healthcare organizations to outwardly project commitment to equity while minimizing changes to organizational practices. For example, an organization could cherry-pick a patient population or the results of analyses to minimize the projected impact of an AI model on health inequities. To address this, there is an opportunity for independent registries that provide transparency and traceability throughout HEAAL procedures to hold healthcare organizations accountable. Similar to the registration of clinical trials, healthcare organizations can register AI product assessments and report progress in conducting HEAAL procedures. Organizations that report outputs that deviate from the initial intended scope of AI product use will face strict scrutiny from internal and external stakeholders.

## Limitations and future directions

While the HEAAL framework is valid, thorough, and user-friendly, it has several limitations. First, the current framework is developed based on the US context. Users seeking to address equity concerns in other countries may encounter gaps in the framework or find that certain procedures are less relevant to their specific contexts.

Second, the framework is not simple; rather, it is highly extensive and detailed. We recognize that its thoroughness might appear intimidating to users. To address this concern, we have provided instructions for procedures in straightforward and plain language. We hope this effort enhances accessibility and promotes better understanding.

Third, the framework was designed and tested using AI products developed in-house. The pediatric sepsis model was built within Duke Health and the two case studies presented at the workshop were also built within NYP and PCCI. To further validate the framework for a broader set of use cases, HEAAL will need to be applied to scenarios where healthcare organizations procure pre-existing AI solutions that are developed externally, which represents the overwhelming proportion of AI implemented in healthcare.

Lastly, HEAAL has not been validated yet for a generative AI use case. By making HEAAL publicly available for organizations to test on their own algorithms, we hope to continue iterating on the framework and adapting it for additional use cases.

## Conclusion

HEAAL comprehensively addresses on all stages of the AI solution lifecycle and draws insights from the perspectives of healthcare delivery organizations and ecosystem partners. Acknowledging the dynamic nature of AI technologies and the evolving landscape of health disparities, we plan to iteratively refine, improve, and adapt HEAAL to ensure that it remains responsive and up to date to emerging health equity issues. Our commitment extends to transparently sharing any updates made on HEAAL through the HAIP website (healthaipartnership.org). With HEAAL, we hope to effectively mitigate health disparities in AI-driven healthcare, while confronting evolving challenges and seizing opportunities. We are dedicated to advancing equitable healthcare delivery and seeking ongoing feedback from practitioners and researchers to stay at the forefront. This collaborative approach invites stakeholders to test HEAAL in practice, provide feedback on its usability, exchange knowledge, and share real-world applications across diverse healthcare settings.

## Supporting information

**S1 Appendix. Overview of Health Equity Across the AI Lifecycle (HEAAL).**
(PDF)

**S2 Appendix. Health Equity Across the AI Lifecycle (HEAAL).**
(PDF)

## Acknowledgments

We thank David Robinson for sharing his insights as a framework developer. We thank Deirdre Mulligan for her support as a Health AI Partnership leader prior to her leave in January 2023. We thank Willie Boag and Shems Saleh for testing a prototype of the framework and providing feedback to improve the clarity and usability of the procedures, especially with a technical focus. We thank Duke Heart Center for helping administer and manage the grant.

## Author Contributions

**Conceptualization:** Jee Young Kim, Alifia Hasan, Alexandra Valladares, Keo Shaw, Danny Tobey, David E. Vidal, Manesh Patel, Inioluwa Deborah Raji, Michael Gao, Suresh Balu, Mark P. Sendak.

**Data curation:** Jee Young Kim, Alifia Hasan, Suresh Balu, Mark P. Sendak.

**Formal analysis:** Jee Young Kim.

**Investigation:** Jee Young Kim, Alifia Hasan, Suresh Balu, Mark P. Sendak.

**Methodology:** Jee Young Kim, Alifia Hasan, Katherine C. Kellogg, William Ratliff, Sara G. Murray, Harini Suresh, Alexandra Valladares.

**Project administration:** Alifia Hasan.

**Supervision:** Suresh Balu, Mark P. Sendak.

**Validation:** William Ratliff, Michael Gao, William Knechtle, Linda Tang.

**Writing – original draft:** Jee Young Kim, Mark P. Sendak.

**Writing – review & editing:** Jee Young Kim, Alifia Hasan, Katherine C. Kellogg, William Ratliff, Sara G. Murray, Harini Suresh, Alexandra Valladares, Keo Shaw, Danny Tobey, David E. Vidal, Mark A. Lifson, Manesh Patel, Inioluwa Deborah Raji, Linda Tang, Suresh Balu.

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
