## [Decision Letter · Decision Letter 0]

19 Jan 2024

PDIG-D-23-00378

Development and preliminary testing of Health Equity Across the AI Lifecycle (HEAAL): A framework for healthcare delivery organizations to mitigate the risk of AI solutions worsening health inequities

PLOS Digital Health

Dear Dr. Kim,

Thank you for submitting your manuscript to PLOS Digital Health. After careful consideration, we feel that it has merit but does not fully meet PLOS Digital Health's publication criteria as it currently stands. Therefore, we invite you to submit a revised version of the manuscript that addresses the points raised during the review process.

Please submit your revised manuscript within 30 days Feb 18 2024 11:59PM. If you will need more time than this to complete your revisions, please reply to this message or contact the journal office at digitalhealth@plos.org. Please include the following items when submitting your revised manuscript:

We look forward to receiving your revised manuscript.

Kind regards,

Gilles Guillot

Academic Editor

PLOS Digital Health

Journal Requirements:

2. Please send a completed 'Competing Interests' statement, including any COIs declared by your co-authors. If you have no competing interests to declare, please state "The authors have declared that no competing interests exist". Otherwise please declare all competing interests beginning with twhe statement "I have read the journal's policy and the authors of this manuscript have the following competing interests:"

3. Please amend your detailed Financial Disclosure statement. This is published with the article. It must therefore be completed in full sentences and contain the exact wording you wish to be published.

b. If any authors received a salary from any of your funders, please state which authors and which funders.

4. We ask that a manuscript source file is provided at Revision. Please upload your manuscript file as a .doc, .docx, .rtf or .tex.

5. Please provide separate figure files in .tif or .eps format only and remove any figures embedded in your manuscript file. Please also ensure that all files are under our size limit of 10MB.

6. We have noticed that you have a list of Supporting Information legends in your manuscript. However, there are no corresponding files uploaded to the submission. Please upload them as separate files with the item type 'Supporting Information'. 

Additional Editor Comments (if provided):

This is a solid and useful piece of work. 

My sole concern is about the choice (also noted by the reviewer) made by the authors to focus the manuscript on a description of the process that led to the framework rather explaining and commenting the framework itself.

The ms makes too much out of the process that lead to the proposed framework (p. 7-12).

The ms would gain interest if the emphasis was placed on the description of the framework itself. 

In particular, the main information brought by this publicaion is contained in Table 1. It should be part of the ms for clarity and informativeness. 

I am afraid this involves a complete rewriting of the paper but this this in order since in its current form, the text sounds more like the minutes of a meeting rather than like a scientific article. 

Minor comments:

Abstract is clear but difference between reliability and validity not clear at this stage.

p.7 "Design researchers collected responses from framework developers and mapped them

on a Miro board, an online whiteboard with infinite canvas, using sticky notes".

The fact that a Miro board was used is not a crucial information. 

It is fully clear to me what design researchers and frmework deveopers are. Please explain. 

p.8 Section "Synthesize key insights of the workshop" is not informative. Please rewrite or remove.

p.8 "Then, design researchers converted the contents on the Miro board

to a single Word document" this information is immaterial. 

Figure 3 is not informative and could be removed.

Table 2 does not seem to be so informative neither to contain any information not contained in Table 5. 

p.15 I think the sentence "To make a distinction between

two scenarios, procedures for evaluating an existing AI solution are written in red and black.

Procedures for evaluating a new AI solution are written in blue and black (Table 2)" actually refers to Table 5.

Data availability statement "Data used in this submission can be accessed in Supporting Information and will be made publicly available on healthaipartnership.org". 

This statement is a bit confusing. What data are you refering to? The Supporting Information document does not seem to contain any "data". Any data intended to be shared should be made available for the review process. 

Please also address carefully the numerous thoughtful comments made by the reviewer.

Reviewers' comments:

Reviewer's Responses to Questions

**Comments to the Author**

1. Does this manuscript meet PLOS Digital Health’s publication criteria? Is the manuscript technically sound, and do the data support the conclusions? The manuscript must describe methodologically and ethically rigorous research with conclusions that are appropriately drawn based on the data presented.

Reviewer #1: Yes

2. Has the statistical analysis been performed appropriately and rigorously?

Reviewer #1: Yes

3. Have the authors made all data underlying the findings in their manuscript fully available (please refer to the Data Availability Statement at the start of the manuscript PDF file)?

Reviewer #1: Yes

4. Is the manuscript presented in an intelligible fashion and written in standard English?

Reviewer #1: Yes

5. Review Comments to the Author

Reviewer #1: First, I think this is an incredibly important topic, albeit a complex one, and still a gap.

The solutioning process and content give every indication of having come from seasoned professionals with a granular understanding of the AI adoption process, as well as AI creation, although this manuscript reads as through the lens of AI adopters. 

I have almost complete confidence in the quality and comprehensiveness of the output, but my concerns are in the complexity of the communication of said output. 

1) Similar recent paper

First, a quick compare and contrast to another recent paper - Abràmoff, M.D., Tarver, M.E., Loyo-Berrios, N. et al. Considerations for addressing bias in artificial intelligence for health equity. npj Digit. Med. 6, 170 (2023). https://doi.org/10.1038/s41746-023-00913-9

Written more through the lens of an AI practitioner, it mapped easier to existing mental frameworks and most of the text was spent understanding examples of what bias means in health AI and how it is introduced. Concept, design, develop, validate, access, monitoring as portrayed on a visual graphic. 

Note: This framework maps more or less to reviewed paper’s buckets, they have chosen to build upon HAIP prior work. Maybe their framework is more intuitive for the intended audience, I cannot judge. The other seems simpler from an MLOps perspective. 

With respect to the reviewed paper, I assume it is aimed at the very same and important problem, but is NOT redundant as the intent seems in providing a very clear roadmap/checklist for the holistic set of health system adoption stakeholders to evaluate up front and monitor equity risks and effects. 

Primarily I mention for the editor, nonetheless, would be good for the authors also to consider the other paper and if any cite or alignment needed as appropriate - it's recent, understandably. Also, suggest to do a quick last check for other recent papers, seems to have become a current topic. 

2) Communication suggestions/simplification

From a pragmatic point of view, the paper is fine. “Here is the high quality checklist of 37 bullets, here is why you should have confidence in the process”. But since it seems the authors understand the importance of onboarding, change management, adoption, etc, I would suggest that adding a few easier to digest aspects to the paper might impact a larger audience, beyond those already determined to implement it. 

2a) (minor) A minor point first, it does seem a particularly high percentage of text goes to the process, and is ordered first. I do believe the process is even valuable in its own right. But is the secondary contribution of the paper, with purpose simply to establish confidence of output. Just be mindful. 

2b) (potentially larger impact but still minor revision) On output communication, for example, I could strongly suggest a simplified visual, mapping 8 process steps to main risks/checks but at a higher level form. (“flag training data for bias: representation, label choice, measurement, etc”) or whatever the authors feel is the most intuitive visual might be. i.e. less words, less bullets, higher level, more visual. 

Again, I make this type of suggestion not to be critical of content, but rather because I feel the addressed problem is important, that the process was well thought out, and hence it would be ideal if there was an easier way for it to be circulated and socialised. 

2c) (minor) Beyond that, but along the same lines, perhaps more words spent on intuition around the 8 buckets and associated risks/steps. Is there any more to be added in granularity or description as far as the ethical metrics, etc. I trust the authors will know better the intended audience and proficiency in topic. Visually, authors could consider description or nick name in small text to Figure 6 ABCD boxes.

3) (food for thought). Lastly, I am pasting in more of a “Health equity for non-experts” read up I have seen recently in smw as I reside in Switzerland, strictly as an example of simple communication. However it will not map 1:1 for the authors purposes but may offer food for thought. 

The pursuit of health equity in the era of artificial intelligence

https://smw.ch/index.php/smw/article/view/3286/5530

https://doi.org/10.57187/smw.2023.40062

Additional thoughts: 

This is touched upon at various points, but an algorithm which leads to an increased inequity in one market could have a significant outcomes boost in another markets, there are different bars for risk-return. The specific examples that come to mind are mostly out of US. 

What does the product mindset for this look like? 

Also it may be instructive to remind the reader that for some tasks, humans are also just predictive models that have their own biases. Although I believe this is captured inherently in the framework.

6. PLOS authors have the option to publish the peer review history of their article (what does this mean?). If published, this will include your full peer review and any attached files.

**Do you want your identity to be public for this peer review?** For information about this choice, including consent withdrawal, please see our Privacy Policy.

Reviewer #1: Yes: Nicholas W. Kelley

---

## [Decision Letter · Decision Letter 1]

28 Mar 2024

Development and preliminary testing of Health Equity Across the AI Lifecycle (HEAAL): A framework for healthcare delivery organizations to mitigate the risk of AI solutions worsening health inequities

PDIG-D-23-00378R1

Dear Dr. Kim,

We are pleased to inform you that your manuscript 'Development and preliminary testing of Health Equity Across the AI Lifecycle (HEAAL): A framework for healthcare delivery organizations to mitigate the risk of AI solutions worsening health inequities' has been provisionally accepted for publication in PLOS Digital Health.

Best regards,

Gilles Guillot

Academic Editor

PLOS Digital Health

Reviewer Comments (if any, and for reference):

Reviewer's Responses to Questions

**Comments to the Author**

1. If the authors have adequately addressed your comments raised in a previous round of review and you feel that this manuscript is now acceptable for publication, you may indicate that here to bypass the “Comments to the Author” section, enter your conflict of interest statement in the “Confidential to Editor” section, and submit your "Accept" recommendation.

Reviewer #1: All comments have been addressed

2. Does this manuscript meet PLOS Digital Health’s publication criteria? Is the manuscript technically sound, and do the data support the conclusions? The manuscript must describe methodologically and ethically rigorous research with conclusions that are appropriately drawn based on the data presented.

Reviewer #1: Yes

3. Has the statistical analysis been performed appropriately and rigorously?

Reviewer #1: N/A

4. Have the authors made all data underlying the findings in their manuscript fully available (please refer to the Data Availability Statement at the start of the manuscript PDF file)?

Reviewer #1: Yes

5. Is the manuscript presented in an intelligible fashion and written in standard English?

Reviewer #1: Yes

6. Review Comments to the Author

Reviewer #1: I personally find the article significantly improved in clarity of communication and hope this will lead to a larger impact. I hope the authors are also happy with the revision.

Figure 4 is great. Results section opening is super helpful.

Structure and procedures section answers the framework points I raised and ‘why’ for some decisions.

I would only have 2 points, neither of which would necessarily need to be addressed. But as reviewers we feel obligated to show we’ve really reread the paper...

First, the couple sections starting page 9 (in my version) starting with “Conducting initial usability testing” and the next 2-3 sections - these could be more concise and higher level. E.g. “By incorporating feedback from data scientists, we ensured X…and/or the final framework became more Y.” Instead it reads as a sequence of things that happened. However, if the authors feel that the current wording would enable others interested in the method (it is after all a methods section) to better learn and implement, then I would say this is fine.

Again, I feel the feedback from the reviewers, including myself, was overall very successfully addressed in this direction.

Lastly, in this sentence, “HEAAL will necessitate 13 significantly higher investment than a process that skips the assessment of health equity impacts”: The word “significantly” might intimidate potential adopters. And that’s fine. Especially since you are making the case for broader support, resources and prioritization. Just be intentional with your words. Is it “significant”, is it “potentially higher”, or perhaps “ginormous”? Or simply an additional bit of insight – 10% more or 10x more? Or just leave it, but you get my point.

Good revisions, makes the act of reviewing feel more valuable.

7. PLOS authors have the option to publish the peer review history of their article (what does this mean?). If published, this will include your full peer review and any attached files.

**Do you want your identity to be public for this peer review?** For information about this choice, including consent withdrawal, please see our Privacy Policy.

Reviewer #1: **Yes: **Nicholas W. Kelley
